# Self-Guided Plan Extraction for Instruction-Following Tasks with Goal-Conditional Reinforcement Learning

## Abstract

We introduce SuperIgor, a framework for instruction-following tasks. Unlike prior methods that rely on predefined subtasks, SuperIgor enables a language model to generate and refine high-level plans through a self-learning mechanism, reducing the need for manual dataset annotation. Our approach involves iterative co-training: an RL agent is trained to follow the generated plans, while the language model adapts and modifies these plans based on RL feedback and preferences. This creates a feedback loop where both the agent and the planner improve jointly. We validate our framework in environments with rich dynamics and stochasticity. Results show that SuperIgor agents adhere to instructions more strictly than baseline methods, while also demonstrating strong generalization to previously unseen instructions.

## 1 Introduction

The instruction-following task (Shridhar et al., 2020; Chevalier-Boisvert et al., 2018; Zhong et al., 2021) involves an AI agent achieving a goal specified as a textual instruction. This task can be framed within reinforcement learning, where the agent must develop a policy to maximize a reward that reflects how well it follows the given instruction. The challenge lies in constructing an optimal policy based on multimodal observations, combining textual and visual information from the environment.

One possible approach to solving the Instruction Following task involves encoding both visual data and textual instructions into a shared latent representation, upon which a policy is subsequently built (Zhong et al., 2019; Lynch et al., 2022; Wang & Narasimhan, 2021). Techniques such as CLIP (Yao et al., 2022) and FiLM (Perez et al., 2018) are commonly used to enhance this multimodal encoding. However, a key limitation of this method arises when the instruction is complex and requires the execution of a lengthy sequence of actions. In partially observable environments or dynamic settings, it becomes particularly challenging for the agent to consistently align the appropriate action with the instruction, especially when faced with diverse and often ambiguous observations.

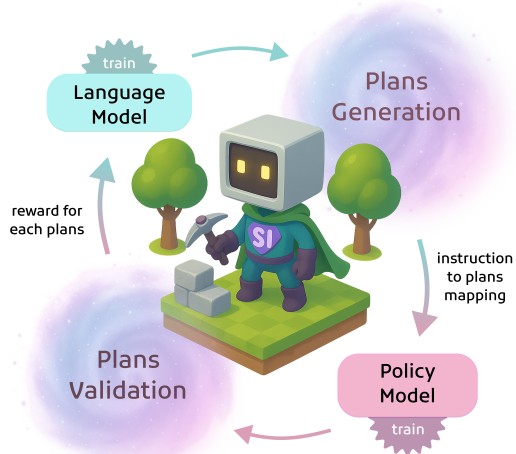

Figure 1: Conceptual diagram of the **SuperIgor** framework designed for Instruction Following.

On the other hand, previous work such as Zhang et al. (2024); Ahn et al. (2022) demonstrate that the instruction-following task can be approached through plan generation, decomposing the instruction into a sequence of high-level actions. In such approaches, a large language model first breaks down the instruction into a structured list of high-level actions. The resulting plan is then encoded into

a structured representation, which can be an embedding obtained from a language model Zhang et al. (2022) or a one-hot vector encoding Volovikova et al. (2024) that is passed to the RL agent for execution. The generated plan, formatted accordingly, is then fed into the RL agent. This approach improves generalization to out-of-distribution tasks, as complex natural language formulations are transformed into a deterministic sequence of steps Wang et al. (2023); Logeswaran et al. (2022); Tan et al. (2024). The primary challenge of such methods is that the set of possible subtasks must be predefined in advance. The agent constructs a plan by selecting from this limited set of tasks, which restricts the method's flexibility when encountering unforeseen situations.

In this paper, we introduce the **SuperIgor** framework for the instruction-following task. Our approach extends the idea of plan generation, where a language model first decomposes an instruction into a structured sequence of subtasks, which is subsequently executed by a reinforcement learning agent. In contrast to prior methods that depend on a fixed set of predefined subtasks, SuperIgor adopts a more flexible strategy by incorporating a self-learning mechanism. Rather than relying on environment-specific datasets to train the language model, our framework enables the model to iteratively refine its plan generation through its own outputs, enhancing generalization to unseen tasks and significantly reducing the need for manual data curation. Furthermore, we demonstrate that SuperIgor performs effectively in dynamic and partially observable environments such as CrafText.

To conclude, our contributions are as follows:

- We propose a new self-supervised training paradigm for the instruction-following task, where high-level plans are generated and refined through interaction between a language model and a reinforcement learning agent—without requiring any manually annotated datasets.

- We introduce a special curriculum to train an RL agent to accurately follow the plan despite sparse reward conditions.

- We implement our approach in the CrafText benchmark and achieve state-of-the-art performance on out-of-distribution tasks, demonstrating the robustness and flexibility of our framework in dynamic and partially observable environments. The dataset and code for SuperIgor are publicly available[1].

## 2 RELATED WORK

**Instruction Follwing Tasks** are formulated differently depending on the type of environment. Construction-centered settings like CraftAssist (Gray et al., 2019) and IGLU (Kiseleva et al., 2022) define the task as building complex 3D structures based on language instructions. Navigation environments such as Touchdown (Chen et al., 2020) and Alfred (Shridhar et al., 2020) focus on guiding an agent through spatial environments or household scenarios using natural language commands. Environments like BabyAI (Chevalier-Boisvert et al., 2018) and HomeGrid (Lin et al., 2023) emphasize planning sequences of basic actions in dynamic, evolving environments conditioned on high-level textual goals. Meanwhile, Messenger (Wang & Narasimhan, 2021) and RTFM (Zhong et al., 2019) present a different formulation: the agent receives textual descriptions of the game's mechanics — such as defining allies, enemies, or victory conditions — and must infer new behaviors by interpreting these dynamically generated rules.

Given the diversity of environments, a variety of approaches to instruction following has been developed, often tailored to the specific task formulation. Among them, the most common strategy is to jointly encode the instruction and the observation, bridging visual and textual modalities. One prominent direction uses shared representation models such as CLIP (Yao et al., 2022), or feature projection techniques like FiLM layers (Perez et al., 2018), to align linguistic and perceptual features Zhong et al. (2019); Paischer et al. (2023); Chevalier-Boisvert et al. (2018). Alternatively, transformer-based architectures, including EmBERT (Suglia et al., 2021) and Vision-and-Language Navigation frameworks (Savva et al., 2019), process multimodal inputs jointly to enhance instruction understanding and execution. Additionally, model-based reinforcement learning approaches, such as Dynalang (Lin et al., 2023), offer an alternative by learning structured policies conditioned on textual goals within dynamic environments.

---

[1] https://anonymous.4open.science/r/SuperIgor-7A4F

**Instruction Following and Planning**. Recent work has shown that large language models (LLMs), when fine-tuned on suitable datasets, are capable of producing detailed and coherent High-Level plans for agents based solely on textual instructions, without relying on visual observations (Jansen, 2020; Zhao et al., 2024; Zhou et al.) (. Building on this capability, a common approach to instruction following is to provide the LLM with the task description, optionally the current environment state, and a structured plan format; the model then generates a sequence of subgoals (e.g., DEPS (Wang et al., 2023), Translated LLM (Huang et al., 2022)). However, such methods have two fundamental limitations. First, the generated subgoals must correspond to a predefined set of skills available in the environment, which requires mapping each subgoal to the closest existing skill using additional heuristics or learned similarity metrics (Logeswaran et al., 2022). Second, these pipelines typically assume the presence of a pre-trained low-level controller that is already capable of executing the predicted skills, leaving the problem of learning a low-level policy unaddressed. Methods that jointly train an RL agent (e.g., SayCan (Ahn et al., 2022), PSL (Wang et al., 2023), or IGOR (Volovikova et al., 2024)) still rely on a predefined skill library or require dense, manually designed reward signals for each subtask. Furthermore, many existing planning systems depend on extremely large LLMs (100B+ parameters), which limits their practicality in resource-constrained settings.

These limitations leave open an important question: how can we learn a low-level policy for instruction following in environments where no predefined set of executable skills is available? In this work, we introduce SuperIgor, a method that addresses this challenge. SuperIgor generates plans without relying on any predefined skill set, learns a low-level policy under sparse rewards (where individual subtask completion cannot be directly verified and reward is given only for accomplishing the full instruction), and adapts the generated plans to support RL training in dynamic and stochastic environments. Importantly, we demonstrate that our method operates effectively using a planning model with only 14B parameters, significantly reducing the computational requirements compared to prior approaches. A detailed comparison of the methods with SuperIgor is presented in the table 3.

## 3 PROBLEM STATEMENT

The environment is formalized as a goal-based Partially Observable Markov Decision Process (POMDP), defined by the tuple $(\mathcal{S}, \mathcal{A}, \mathcal{O}, \mathcal{T}, \mathcal{R}, \mathcal{G}, \gamma)$. The agent receives a natural language instruction $I$ and must achieve the corresponding latent goal $g \in \mathcal{G}$. Each observation $o \in \mathcal{O}$ contains partial information about both the environment and the instruction $I$. The agent learns a grounding function $f_g(I)$ to infer the latent goal $g = f_g(I)$.

The policy $\pi(a \mid o)$ selects actions based on observations to maximize the expected cumulative reward: $\pi^* = \arg\max_\pi \mathbb{E}_\pi \left[ \sum_{t=0}^{T} \gamma^t R(s_t, a_t, g) \,\middle|\, o_0 \right]$. The environment involves stochastic transitions $\mathcal{T}(s' \mid s, a)$ and partial observability, requiring the agent to infer goals and act effectively under uncertainty.

We extend this setup by **introducing plans**. In the planning-augmented formulation, the agent does not receive the instruction $I$ directly. Instead, it is provided with a plan $p = (p_1, p_2, \ldots, p_n)$ derived from $I$, where each step $p_i$ corresponds to an intermediate subgoal $g_i = f_g(p_i)$. At each timestep, the agent observes the environment together with the current plan step $p_{\phi(t)}$. The optimization objective becomes: $\pi^* = \arg\max_\pi \mathbb{E}_\pi \left[ \sum_{t=0}^{T} \gamma^t R\left(s_t, a_t, g_{\phi(t)}\right) \,\middle|\, o_0 \right]$, where $g_{\phi(t)}$ is the subgoal associated with the active plan step.

In contrast to settings with predefined subtasks and explicit intermediate rewards, our formulation introduces two key challenges:

1. **Subtask alignment under sparse rewards.** The agent must discover how its behavior aligns with intermediate subgoals despite only receiving sparse, delayed feedback upon completing the full instruction. This exacerbates the credit assignment problem.

2. **Extended action space.** The agent must also decide when to terminate the current subtask. This requires augmenting the action space with control operations (e.g., a *DONE* action), which increases both exploration complexity and the difficulty of learning effective switching strategies.

# 4 SUPER IGOR

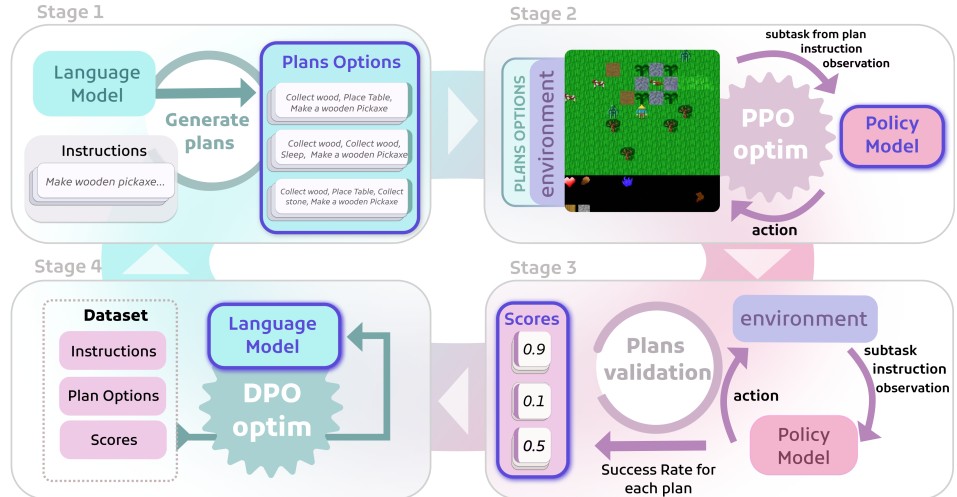

Figure 2: Super Igor Pipeline: The pipeline consists of four stages: (1) a language model generates multiple plan options for a given instruction; (2) a policy model is trained via PPO to execute each plan in the environment; (3) each plan is validated by measuring its execution success rate; (4) the language model is optimized using Direct Preference Optimization (DPO) based on plan performance scores. This iterative loop refines both plan generation and execution.

Super Igor framework proposes a method for jointly training a large language model and a reinforcement learning agent to solve instruction-following tasks. The LLM is responsible for transforming natural language instructions into structured plans, i.e. sequences of subtasks. The RL agent learns to execute these plans in the environment by interacting with it and maximizing delayed rewards.

The training process proceeds through the following stages:

1. **Plan Generation** (4.1): The LLM extracts possible subtasks from instructions and generates multiple candidate plans in natural language during the initial cycle (Cycle 1). In subsequent cycles (Cycle 2–N), the candidate pool is iteratively refined by filtering and re-prioritization, based on how well the plans align with the RL agent's performance.

2. **Policy Learning** (4.2): The RL agent is trained to execute the selected plans in the environment.

3. **Plan Validation** (4.3): The quality of candidate plans is evaluated according to the RL agent's success rate and execution trajectories.

4. **LLM Fine-Tuning** (4.4): The language model is fine-tuned with feedback derived from validation, aligning its scoring of plans with the agent's actual performance.

## 4.1 PLAN GENERATION

In our approach, we first generate all possible plans for the training set in zero-shot mode during the initial cycle. In subsequent cycles, we progressively reduce the set of candidate plans by filtering out those that perform poorly for the agent. Concretely, the initial cycle produces the complete pool of plans, while later cycles re-prioritize them using the LLM's negative log-likelihood (NLL) score. Importantly, we leverage the agent's performance feedback as a preference signal to fine-tune the LLM with DPO, so that the model learns to align its scoring with the agent's actual success in executing the plans.

**Zero-shot plans candidates generation (Cycle 1).** Since the language model used for plan generation may not fully capture the exact dependencies and interaction rules of the target environment, we propose a structured procedure that separates the identification of goals from the reasoning about prerequisite constraints. The method unfolds in four steps.

First, we build a *subtask base* by extracting and canonicalizing possible subtasks from the instruction dataset, creating a unified vocabulary that reduces synonymy and ensures consistency. Each subtask is expressed in natural language, but in a strict normalized format that allows passing them one-by-one to the policy without ambiguity.

Second, the model generates a *goal-level plan*, producing for each instruction a single conceptual representation of its intended outcome, expressed in terms of the established subtask base. This step abstracts away from concrete execution details and captures only the high-level intent.

Third, we induce a *subtask ontology* that encodes the model's hypotheses about prerequisite relations, i.e., which subtasks must be completed before others can be attempted. This provides a structured view of dependencies across the subtask base.

Finally, we perform *plan expansion*, where the single conceptual plan is unfolded into multiple detailed plans, with their number corresponding to the hypotheses proposed by the model. The ontology ensures that these expanded variants remain consistent with prerequisite relations and avoid contradictions.

This approach provides two key benefits. First, it improves plan consistency by constructing plans from a shared set of subtasks and their relations, rather than from independent and potentially contradictory structures. Second, it supports partial normalization, since the model, when processing new instructions, tends to reuse previously identified subtasks, thereby reducing the proliferation of synonymous formulations. The details of the method and pseudocode are provided in Appendix A, and the prompts are presented in Appendix B.

**Plans re-prioritizing for RL-agent (Cycles 2-N).** After obtaining the initial feedback on agent performance for the generated plans and applying LLM fine-tuning (Subsection 4.4), subsequent cycles focus on re-prioritizing the candidate set. In each cycle, plans are rescored using the language model's negative log-likelihood (NLL), which reflects how natural or plausible a plan is according to the model. Plans are then ranked by this score, and only the top-performing subset is retained for further training. As cycles progress, this iterative filtering process gradually narrows the candidate space, aligning the remaining plans both with the agent's empirical success and with the model's learned preferences.

## 4.2 POLICY LEARNING

After the plans have been generated, we train a reinforcement learning agent using the step-wise plan observation setting (Subsection 3). At each timestep, the agent observes the environment and receives an embedding of the current plan step. It must learn to align actions with plan steps based on a delayed reward signal provided only upon successful completion of the entire plan. We use the PPO algorithm to train the policy.

To address the sparse reward problem in training, we introduce **Skill Curriculum Learning**. The core principle is to create a dynamic curriculum that begins with the simplest single-subtask tasks, allowing the agent to learn foundational behaviors under a relatively dense reward signal.

As the agent trains, we monitor its Success Rate (SR) for each subtask. Once a subtask's SR surpasses a predefined threshold $\tau$, it is marked as "mastered." This mastery triggers an update to the curriculum: the set of active training plans is expanded to include any plan composed of already mastered subtasks and, at most, one new, un-

---

**Algorithm 1** Skill Curriculum Learning

**Require:** Set of all plans $\mathcal{P}$, success-rate threshold $\tau$
1: Initialize mastered skills $\mathcal{M} \leftarrow \emptyset$
2: Initialize PPO agent $\pi_\theta$
3: Initialize active plans

$$\mathcal{S} \leftarrow \{p \in \mathcal{P} \mid p \text{ contains exactly one skill}\}$$

4: **while** training not converged **do**
5:     Train $\pi_\theta$ on active plans $\mathcal{S}$ and collect rollouts
6:     For each skill $s$, compute success rate:

$$SR(s) = \frac{\text{\# Successful episodes containing } s}{\text{\# Total episodes containing } s}$$

7:     **if** $SR(s) \geq \tau$ **then**
8:         Add to mastered skills $\mathcal{M} \leftarrow \mathcal{M} \cup \{s\}$
9:     **end if**
10:    Update plans

$$\mathcal{S} \leftarrow \{p \in \mathcal{P} \mid p \text{ has at most one unmastered skill}\}$$

11: **end while**
12: **return** $\pi_\theta, \mathcal{M}$

mastered subtask. This incremental expansion, detailed in Algorithm 1, ensures a smooth learning gradient and prevents the agent from being overwhelmed.

## 4.3 PLAN VALIDATION

To evaluate the quality of each proposed plan, we repeatedly execute the RL agent in the environment using that specific plan as input. This process is essential due to the highly dynamic and stochastic nature of the environment, where outcomes can vary significantly across runs even for the same plan and initial instruction.

As a result, a single rollout is not sufficient to reliably assess plan effectiveness. Instead, we aggregate statistics over multiple rollouts, such as the average success rate or reward, to obtain a more stable and interpretable estimate of how well the plan supports instruction completion. This repeated evaluation allows us to more confidently associate a given plan with its empirical performance and to use this signal to guide future training and model selection.

## 4.4 LLM FINE-TUNING

In the first cycle, we warm-start the language model by supevised finetuning (SFT) to reproduce the same plans that were obtained during the zero-shot generation stage (see Section 4.1). This step adapts the model to the specific distribution of plans relevant to the target environment, ensuring better alignment with the initial candidate pool.

After this supervised adaptation, subsequent cycles incorporate plan-level quality signals collected during execution and validation. These signals capture how well individual plans support the agent in solving the target task. Based on them, we construct a dataset of plan pairs with explicit preferences—each pair contains a higher-scoring (preferred) and a lower-scoring (non-preferred) candidate. This preference dataset is then used to fine-tune the model with DPO, allowing the LLM to internalize the agent's feedback and improve its plan generation over time.

Importantly, the DPO signal in our framework serves as a lightweight plan-selection bias rather than a precise credit assignment mechanism. During early learning, the RL agent naturally makes progress on some plan structures more easily than others. DPO increases the probability of these early-learnable plans, effectively forming an automated curriculum over plan decompositions. Plans that produce no early progress are not labeled as incorrect; they are simply deprioritized because the agent is not yet able to learn from them effectively. This approach intentionally sidesteps the challenge of precisely attributing failures and instead focuses on accelerating training by reinforcing empirically useful plan patterns.

# 5 EXPERIMENTS

In this section, we describe the experiments conducted to answer the following research questions (RQ):

**RQ1. (Effectiveness and Generalization of Auto-Generated Plans)**: How well can the SuperIgor agent learn to follow instructions by leveraging LLM-generated plans, and how well does this learned behavior generalize to new instructions? We measure effectiveness as the agent's final success rate on training tasks (Atomic and Combo splits). We measure generalization using final success rates on two test sets: Paraphrases (same goals, new wording) and New Objects (new goal combinations).

**RQ2. (Policy Training under Sparse Feedback)**: How well can the SuperIgor policy model be trained to follow plans under sparse feedback? The primary metric for this is the final SR on the training tasks.

**RQ3. (Agent Effectiveness with Iterative SuperIgor Cycles)**: How does the agent's performance evolve over multiple iterations of the SuperIgor planning-training cycle?

## 5.1 ENVIRONMENT.

We conduct our experiments in the CrafText benchmark (Volovikova et al., 2025), which provides a unified testbed for evaluating instruction-following agents in multimodal, dynamic, and partially observable environments. It enables us to assess both the agent's ability to interpret diverse linguistic formulations and to adapt to novel goals. The world of CrafText closely resembles Minecraft, with episodes varying due to autonomous stochastic mob behavior, randomized resource distribution, and asynchronous events. Moreover, the agent must manage survival constraints such as hunger, thirst, and hostile entities, introducing competing objectives beyond mere instruction completion. Importantly, the conditions under which the agent must execute instructions change across episodes, further increasing the complexity of the setting.

We use the EASY split of the dataset, which contains over 900 instructions and a vocabulary of more than 1,500 unique words. The dataset is structured to rigorously test different aspects of learning and generalization. The training set consists of two types of instructions: **Atomic**, which specify a single, indivisible goal (e.g., "craft a furnace"), and **Combo**, which combine multiple atomic goals into a sequence (e.g., "craft a furnace and then collect wood").

To evaluate the agent's ability to generalize, the evaluation protocol employs two distinct test sets. The **Paraphrases** set contains Combo instructions from the training set (the same goals as in Combo) reformulated with novel vocabulary and syntax, testing robustness to linguistic variation. The **New Objects** set introduces new combinations of atomic goals that appeared during training but never occurred together in a single instruction, directly testing compositional generalization. The sizes of the Combo / Paraphrases / New Objects splits are comparable.

In this dataset, task composition often involves overlapping subtasks. For example, crafting a furnace first requires making a wooden pickaxe and collecting stone—the same steps needed to craft a stone pickaxe or to smelt metal. As a result, agents may learn to rely on broadly useful routines that solve many tasks without attending to the instruction itself. This undermines the central objective of instruction-conditioned learning: instead of interpreting language, agents simply optimize reward by executing generic behavioral patterns. To prevent such behavior, we apply a strict interaction protocol: an instruction is marked as successful only when all of its goals are fully and precisely completed, with no extraneous steps added. To distinguish this challenging setting from the standard benchmark, we refer to it as EASY-STRICT in our experiments. Further details regarding the environment and instruction examples are provided in Appendix D.

## 5.2 EXPERIMENTS SETUP

In our pipeline, we generate plans using Qwen2.5-14B-Instruct[2], fine-tune it for one epoch with DPO ( $\beta = 0.5, \text{lr} = 1 \times 10^{-5}$ ) to stabilize local updates, and then train policies with PPO-T (lr $= 0.001, \varepsilon = 0.02$) and Skill Curriculum Learning for 2.5B steps. We validate by executing 10 plans across 50 seeds to assess robustness. Two full cycles were conducted, with evaluations before and after LLM fine-tuning, and results compared against baselines at 2.5B and 5B steps (Figure 3). Additional hyperparameters are described in more detail in Appendix N.

## 5.3 BASELINES

For our comparative analysis, we use several established baselines from the original CrafText study (Volovikova et al., 2025). PPO-T (Text-Augmented PPO) augments PPO with textual grounding: instructions are encoded using a frozen DistilBERT [CLS] embedding, concatenated with CNN-based visual features, and processed by a GRU to maintain temporal context. PPO-T+ (Plan-Augmented PPO) extends this by first translating each instruction into a structured plan with GPT-4, and then providing the agent with a plan embedding instead of the raw instruction.

FiLM (Perez et al., 2018) offers an alternative integration of language and vision. Here, instruction embeddings generate parameters that modulate CNN outputs via Feature-wise Linear Modulation layers, allowing textual context to directly shape visual feature processing.

---

[2]https://huggingface.co/Qwen/Qwen2.5-14B-Instruct

To ensure consistency, all baselines follow a strict protocol requiring the DONE action to signal task completion, with success only counted when both the instruction is satisfied and DONE invoked. We also evaluate an *Auto-DONE* (*Soft-*) variant, where episodes terminate automatically upon completion, and include an Oracle agent trained with PPO-T and Skill Curriculum Learning on human-written ground-truth plans.

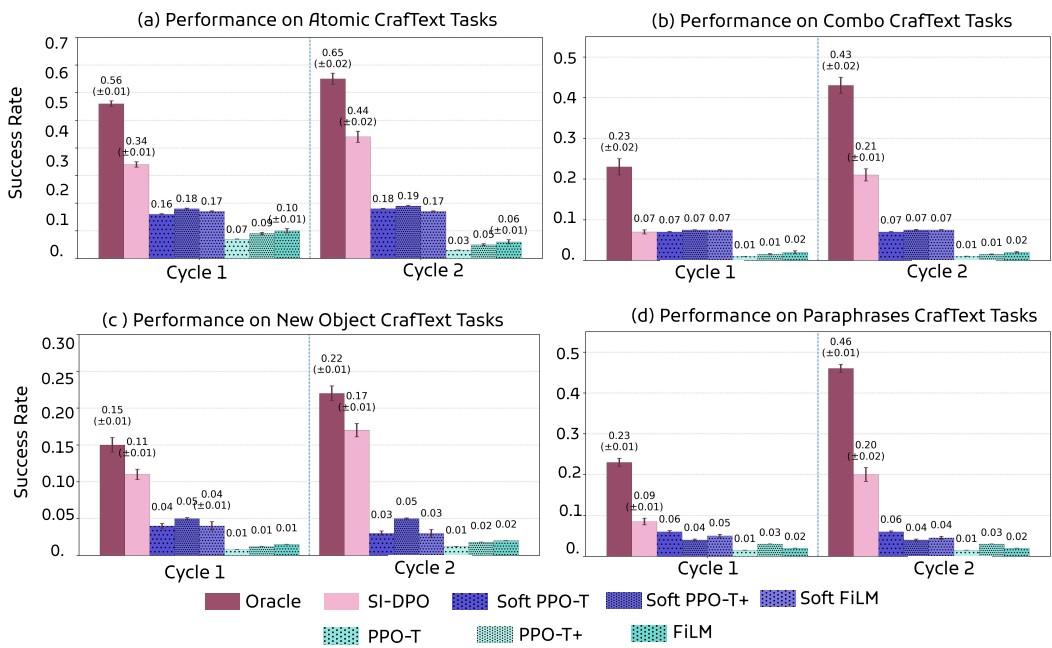

Figure 3: Comparison of SuperIgor and baseline performance on CrafText tasks (Atomic / Combo / New Objects / Paraphrases). SI-SFT denotes SuperIgor validated on plans generated after LLM supervised fine-tuning, while SI-DPO denotes SuperIgor validated after LLM DPO fine-tuning. All agents were evaluated at 2.5 billion steps (corresponding to the first cycle in the SuperIgor approach) and 5 billion steps (corresponding to the second cycle).

## 5.4 EXPERIMENTS RESULT

**RQ1. Effectiveness and Generalization of Auto-Generated Plans in the SuperIgor Pipeline**

**a) Auto-generated plans train agents far more effectively than instruction-only baselines.** On Atomic tasks (Figure 3(a)), SuperIgor agents (SI-DPO / SI-SFT) reach 0.35–0.45, compared to only 0.10–0.19 for instruction-only RL baselines. Oracle remains higher at 0.56–0.65, but the SuperIgor → Oracle gap ($\approx 0.20$) is much smaller than the Baselines → SuperIgor gap ($\approx 0.25$–0.30), clearly showing the value of plan supervision. On Combo tasks (Figure 3(b)), SuperIgor achieves 0.21, outperforming baselines at 0.08, while Oracle reaches 0.46. The wider gap to Oracle here can be explained by the fact that SI agents must simultaneously learn up to 20 alternative plans, whereas Oracle is trained on a single expert-aligned plan, which simplifies optimization.

**b) Agents trained with auto-generated plans generalize on unseen goals better than those trained with Oracle plans.**

On Combo tasks, Oracle achieves 0.46, while SuperIgor reaches 0.21. But on New Object tasks (Figure 3(c)), Oracle drops sharply to $\approx 0.22$, while SI decreases more moderately to 0.12–0.17. Thus, although SI lags in absolute terms, its performance is more stable: the Oracle–SI gap shrinks from 0.25 on Combo to only 0.05–0.10 on New Object tasks. We attribute this stronger generalization precisely to the fact that SI agents learn from multiple alternative plans per instruction, which exposes them to richer variability during training.

**c) Agents trained with auto-generated plans do not lose performance when instructions are paraphrased.**

Paraphrases reuse (Figure 3 (d)) the same goals as in Combo tasks but are expressed in different linguistic forms. In Cycle 1, SI-DPO performance increases from 0.07 on Combo to 0.09 on Paraphrases. In Cycle 2, SI-DPO remains stable, with 0.21 on Combo and 0.20 on Paraphrases. This shows that SuperIgor agents can successfully transfer their learned strategies to differently worded instructions, maintaining performance even when the language of the goal changes.

**RQ2. Policy Training under Sparse Feedback**

**a) Skill Curriculum Learning enhances agent to learn more subtasks compared to unstructured training**

We evaluate the training process by the number of unique subtasks the agent masters over time. A subtask is considered "mastered" once its success rate surpasses a 70% threshold. This metric provides a clearer insight into the agent's growing capabilities and its ability to handle compositional tasks. We compare three configurations, with the results visualized in Figure 4.

The agent trained with **Skill Curriculum on Oracle Plans** sets a practical upper bound for performance. By the 10 billion step mark, it successfully masters **14 distinct subtasks**. It signifies that the agent has acquired almost the entire 'mining' technology tree: all the achievements from collecting wood to collecting iron. Furthermore, it demonstrates the ability to execute complex, combined instructions that require interleaving subtasks from different progression branches, such as eating, drinking, and collecting resources within a single, coherent plan.

Agent trained **on Oracle plans without the Skill Curriculum** perform worse with only mastered **5 basic subtasks**. Even with a flawless plan, the agent fails to learn without a structured progression that allows it to build foundational skills first. This finding confirms that Skill Curriculum helps to overcome sparse feedback problem and enables agent abilities to learn more subtasks.

**b) SI-Initial plans are a good initial approximation of optimal plans**

**Skill Curriculum with SI-Initial plans** graph follows this trajectory closely, mastering **12 subtasks** within the same timeframe. This demonstrates the high quality of our SI-INITIAL plan generation, as it enables the agent to acquire most of the subtasks achievable even with perfect plans. The gap between these two curves represents the remaining challenge in our automated plan generation.

In conclusion, the curriculum is not just beneficial, it is *critical* for meaningful skill acquisition in this environment. The ablation clearly shows that our Skill Curriculum Learning framework is the key enabler of learning, while our SI-INITIAL procedure generates plans of sufficient quality to unlock a significant portion of the agent's potential and a good baseline for futhermore plan generation improvement using SuperIgor framework.

**RQ3. Agent Effectiveness with Iterative SuperIgor Cycles**

**a) Plan-following quality improves across cycles.** On Atomic tasks (training, Figure 3, (a)), SI-DPO increases from 0.34 in Cycle 1 to 0.43 in Cycle 2. On Combo tasks (training, Figure 3, (b)), SI-DPO grows from 0.06 in Cycle 1 to $\approx$ 0.21 in Cycle 2. On New Object tasks (testing, Figure 3, (c)), SI-DPO declines only slightly from $\approx$ 0.21 to 0.12–0.17, showing that performance improves with additional SuperIgor cycles on both training and testing setups and remains relatively stable when moving to unseen goals.

**b) Plan reprioritization under DPO illustrates the process by which language models are incrementally grounded in the agent's**

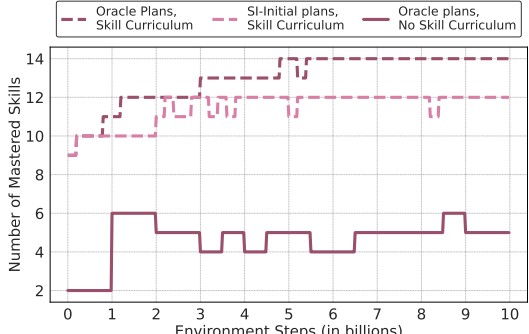

Figure 4: A comparative analysis of the number of mastered subtasks over 10 billion environment steps. The results highlight the critical role of the Skill Curriculum, as agents trained without it fail to learn, even with optimal Oracle Plans.

**behavior and the underlying environment mechanics..** The re-ranking visualization (Appendix F, Figure 8) shows how plans shift across SFT, DPO-C1, and DPO-C2. Success Rates range from 0.68 to 0.86. A plan with SR = 0.86 steadily climbs to the top across cycles, while weaker plans

with SR $\approx 0.68$ remain consistently at the bottom. These changes are gradual rather than abrupt, suggesting that DPO provides a soft grounding signal that progressively aligns plan priorities with the agent's execution success. Exposure to multiple alternative plans per instruction during training enriches variability, which explains why SI agents, although weaker in performance, generalize better than Oracle on unseen tasks.

# 6 CONCLUSION

In this work, we introduced SUPERIGOR, a novel framework that teaches agents to follow complex instructions in sparse-reward environments by iteratively aligning an LLM planner with an RL policy using agent feedback. Our experiments lead to several key conclusions.

First, our core contribution—the iterative alignment of plans using DPO—is highly effective. The SUPERIGOR framework improves both plan quality and agent performance across training cycles by providing a soft grounding signal that progressively aligns the LLM's preferences with the agent's real-world execution capabilities.

Second, we find that a structured curriculum is essential. Our experiments revealed that even with perfect, human-authored Oracle Plans, the agent fails to learn complex subtasks. Our Skill Curriculum Learning framework solves this by enabling the agent to master foundational skills first, demonstrating that managing task complexity is as crucial as providing a correct plan.

Finally, our work revealed that agents trained on a single, optimal Oracle Plan generalize poorly to unseen goal combinations. In contrast, agents trained on the diverse set of auto-gengerated plans from SUPERIGOR exhibit far more robust generalization. This suggests that exposure to a varied set of "good-enough" plans is more beneficial for developing flexible policies than training on a single, narrow path to success.

## REPRODUCIBILITY STATEMENT

The learning process is described in detail in the section 5.2. The hyperparameters are shown in the Appendix N. The computing resources used for conducting experiments are described in the section J. The full code base is available for download to ensure reproducibility of the results, the link is in the Introduction section.

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

APPENDIX

## A  PLANS GENERATION:CANDIDATES GENERATION

In or plan extraction method the goal is to elicit the model's hypotheses about the dependencies between subtasks in the environment. In the first step we construct a subtask bank $B$, i.e., the set of all candidate subtasks derived from the instruction set. For each instruction $I \in \mathcal{D}$, we prompt the language model $f_{\text{LLM}}$ to generate a goals plan $\mathcal{P}[I]$, i.e., the set of goal subtasks directly required by the instruction. The model is provided with the current contents of the subtask bank $B$, which encourages reuse of already known subtasks and reduces the introduction of redundant synonyms. If the generated goals contain subtasks not yet present in $B$, they are added. At the initial iteration the bank is empty, so all subtasks generated by the model are included. The complete process is summarized in Algorithm 2.

Once a sufficiently rich subtask bank $B$ has been established, ontological dependencies between subtasks are extracted. For each target subtask $t \in B$, the language model is queried multiple times to determine which elements from $B$ are required for the completion of $t$. For every candidate dependency $(r \to t)$, its probability is estimated as

$$P(r \to t) = \frac{k_t}{N},$$

where $k_t$ denotes the number of times subtask $r$ was identified as necessary for $t$ and $N$ is the number of queries. To filter out spurious associations, the Wilson confidence interval is applied to the resulting probabilities. The procedure is carried out in two passes: first over the entire bank $B$, and then restricted to the subtasks previously identified as relevant, which refines the weighting of relations. The final output is an ontology graph $G = (V, E)$ that encodes the model's hypothesized structure of interrelations among subtasks. The full procedure is summarized in Algorithm 3.

After constructing the ontology $G = (V, E)$, each goal plan $\mathcal{P}[I]$ is expanded with its dependencies. For every subtask $s \in \mathcal{P}[I]$, we recursively collect all prerequisites in $G$. The union of these subtasks with the original goals defines the plan's vertices, which are then topologically sorted so that prerequisites precede dependents. The result is a linearized plan $P$ containing the goals and all supporting subtasks (Algorithm 4).

---

**Algorithm 2** Subtask Bank Update

**Require:** Instruction stream $\mathcal{D}$, language model $f_{\text{LLM}}$
**Ensure:** Subtask bank $B$, goals plans $\mathcal{P}$
 1: Initialize subtask bank $B \leftarrow \emptyset$
 2: Initialize goals plans $\mathcal{P} \leftarrow \emptyset$
 3: **for** each instruction $I \in \mathcal{D}$ **do**
 4:    Identify goal subtasks conditioned on $B$:

$$S \leftarrow f_{\text{LLM}}(I, B)$$

 5:    **for** each subtask $s \in S$ **do**
 6:      **if** $s \notin B$ **then**
 7:        $B \leftarrow B \cup \{s\}$
 8:      **end if**
 9:    **end for**
10:    Goals plan for $I$: $\mathcal{P}[I] \leftarrow S$
11: **end for**
12: **return** $B, \mathcal{P}$

---

**Algorithm 3** Ontology Construction

**Require:** Subtask bank $B$, language model $f_{\text{LLM}}$, queries per pass $N$, threshold $\tau$
**Ensure:** Ontology graph $G = (V, E)$
 1: Initialize counts $count(r, t) \leftarrow 0$ for all $r, t \in B, r \neq t$
 2: **for** each target subtask $t \in B$ **do**
 3:    **for** two passes **do**
 4:      Define candidate set $C$:

$$C \leftarrow \begin{cases} B \setminus \{t\}, & \text{pass 1} \\ \{r \in B : count(r, t) > 0\}, & \text{pass 2} \end{cases}$$

 5:      **for** $i = 1 \ldots N$ **do**
 6:        Query prerequisites:

$$R \leftarrow f_{\text{LLM}}(t, C)$$

 7:        **for** each $r \in R$ **do**
 8:          $count(r, t) \leftarrow count(r, t) + 1$
 9:        **end for**
10:      **end for**
11:    **end for**
12: **end for**
13: Initialize edge set $E \leftarrow \emptyset$
14: **for** each pair $(r, t)$ **do**
15:    Compute probability:

$$\hat{p}(r \to t) = \frac{count(r, t)}{N}$$

16:    Compute Wilson lower bound $LB(\hat{p}, N)$
17:    **if** $LB \geq \tau$ **then**
18:      $E \leftarrow E \cup \{(r \to t)\}$
19:    **end if**
20: **end for**
21: **return** $G = (V = B, E)$

---

**Algorithm 4** Final Plan Generation from Ontology

---

**Require:** Instruction $I$, goals mapping $\mathcal{G}$, goals plan $\mathcal{P}$, ontology $G = (V, E)$
**Ensure:** Final plan $P$
 1: Retrieve goal subtasks: $S \leftarrow \mathcal{G}[I]$
 2: Initialize plan vertex set: $U \leftarrow S$
 3: **for** each $s \in S$ **do**
 4:    Expand prerequisites via ontology:

$$D \leftarrow \text{PREREQCLOSURE}(s, G)$$

 5:    $U \leftarrow U \cup D$
 6: **end for**
 7: Extract induced subgraph: $G_U \leftarrow G[U]$
 8: Topologically sort $G_U$ to obtain ordered plan $P$
 9: **return** $P$

---

## B   PLANS GENERATION: PROMPT FOR PLAN GENERATION

```
You control an agent in a 2D game with simplified Minecraft environment.
You will need to provide a detailed step-by-step plan for following the user's instructions.
You must include all the preliminary steps that it needs to complete.

You are controlling an agent in a 2D game set within a simplified Minecraft-like environment.
The agent starts from scratch with an empty inventory and no gathered resources.
Your task is to generate a step-by-step plan that enables the agent to follow a given user instruction.

What you must do:
- Break down the instruction into atomic actions the agent needs to perform.
- Include all necessary preliminary steps, such as gathering or crafting resources.
- Assume the agent has nothing at the beginning | you must plan from the ground up.
- Output your answer as a Python list of strings.
- Each string must represent one atomic skill invocation, written on a separate line.

Format for each step:
"skill_name(arg1 = value1, arg2 = value2, ...)"
- skill_name: the name of the primitive skill or action the agent will execute.
- Inside the parentheses, list all required arguments with their names and corresponding values.

Example:
gather_resource(resource_type = wood)

Each of the step agents will be implemented without knowledge of what it did before,
so it can only rely on observation and the current step.
Therefore, each step must be self-sufficient and not require knowledge of past steps.

"If the instruction doesn't specify what the agent needs to do and is more general|like
'Explore the world' or 'Go out and examine the world around you'|send explore(object=world).
In this case, the plan should consist of only one step: "explore(object=world)"."

Send your answer as a python list.
Instruction: Make a pickaxe from wood
Answer:
["gather_resource(resource_type = wood)",
"gather_resource(resource_type = wood)",
"create_item(item_type = table)",
"gather_resource(resource_type = wood)",
"gather_resource(resource_type = wood)",
"create_item(item_type = wooden_pickaxe)"]

Send your answer as a python list.
Instruction: $INSTRUCTION$
Answer:
```

## C   ADDITIONAL EXPERIMENTS

### C.1   ABLATION STUDY: SUPERIGOR FRAMEWORK

To quantify the contribution of each module of the SuperIgor framework, we conduct an ablation study in which individual components are removed from the training pipeline. We evaluate the influence of four factors: (1) Ontology-Based Training Plan Generation, (2) Curriculum design in the RL stage, (3) LLM plan-model pretraining (SFT), and (4) DPO finetuning based on RL agent performance signals. Table 1 presents the results of this experiment, where we measure the SuperIgor agent's SuccessRate on the Atom subset of the CrafText instruction dataset. The analysis of the results yields two central findings.

Table 1: Ablation study of the SuperIgor framework, measuring agent SuccessRate on the Atom subset of the CrafText dataset across two training cycles

| Ontology | Curriculum | DPO | SFT | Cycle-1 | Cycle-2 |
|:---:|:---:|:---:|:---:|:---:|:---:|
| ✗ | ✓ | ✓ | ✓ | 0.06 | N/A |
| ✓ | ✗ | ✓ | ✓ | 0.08 | N/A |
| ✓ | ✓ | ✗ | ✓ | 0.34 | 0.39 |
| ✓ | ✓ | ✓ | ✗ | 0.25 | 0.13 |
| ✓ | ✓ | ✓ | ✓ | **0.35** | **0.45** |

**(1) Curriculum is effective only when paired with high-quality, ontology-structured plans.** Although a full-cycle evaluation may give the impression that the primary gains come from curriculum learning, the results of this ablation study show that its effectiveness emerges only in combination with ontology-guided plan generation. Without ontology (i.e., without structured, hierarchical plans), the curriculum has no meaningful ordering signal and fails to provide improvement: Cycle-1 performance drops to $0.06$ when ontology is removed.

Ontology-based plans, however, naturally encode a hierarchy of instructions and goals, enabling a principled progression from simpler to more complex targets. This hierarchical structure is precisely what makes a curriculum implementable: the RL agent can first master low-complexity goals and then gradually advance to more difficult ones. When ontology is present, this alignment between plan structure and staged learning produces large gains, improving Cycle-1 performance from $0.06$ (no curriculum) to $0.35$ (with curriculum).

**(2) DPO improves the RL agent by learning to prioritize plans that lead to higher-quality behavior.** Unlike SFT, which is trained to reproduce the ontology-induced distribution of plans, DPO directly leverages RL performance as a preference signal: it learns to rank plans higher when they empirically yield better agent behavior. Removing DPO results in weaker prioritization: the RL agent reaches only $0.39$ in Cycle-2 without DPO, compared to $0.45$ when DPO is included. Thus, DPO systematically shifts the plan distribution toward behaviorally effective plans, accelerating and amplifying the RL agent's improvement across cycles.

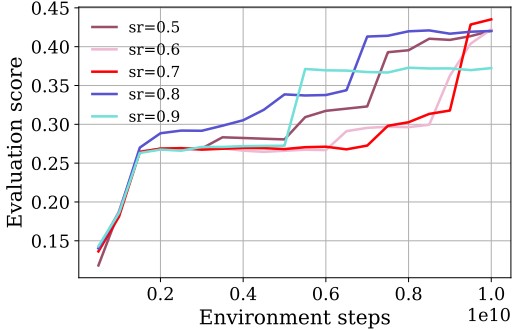

Figure 5: Ablation of the skill-mastery threshold $\tau$. The plot shows evaluation scores on the Atomic and Combo tasks during training for different $\tau$ values.

### C.2   ABLATION STUDY: SKILL MASTERY THRESHOLD

We conducted an ablation study to analyze the sensitivity of the Skill Curriculum Learning to the mastery threshold parameter $\tau$. Figure 5 presents the final performance of the Skill Curriculum

Learning agent after 10 billion environment steps in CrafText-Symbolic configuration for different values of $\tau$ ranging from $0.5$ to $0.9$.

The results demonstrate that $\tau = 0.7$ provides an optimal balance for curriculum progression. We hypothesize that lower thresholds ($\tau = 0.5$) allow the agent to progress too quickly to complex skills before achieving reliable proficiency, while higher thresholds ($\tau = 0.9$) cause the agent to spend excessive time perfecting basic skills, slowing overall learning. The $\tau = 0.7$ value strikes an optimal balance between progression speed and skill reliability.

### C.3 Ablation Study: Choice of LLM for Ontology and Training-Plan Generation

To understand how the choice of language model affects the quality of the ontology and the generated training plans we conducted an ablation study comparing several families of LLMs. For each model, we regenerated both the ontology and the full training dataset (plans), and then trained an RL agent using our Skill Curriculum Learning procedure.

Table 2 reports the agent's success rate on the training split under different planner models. The experiment includes models from the Qwen and Gemma families, as well as the larger `microsoft/NextCoder-32B` model.

Table 2: Ablation on the choice of LLM used for generating both ontology and training plans. We report success rate on the training set.

| LLM | Qwen1.5-32B | NextCoder-32B | Qwen1.5-14B | Gemma-12B | Qwen-7B |
|---|---|---|---|---|---|
| **SR (Train)** | 0.43 | 0.26 | 0.35 | 0.14 | 0.22 |

**(1) Larger models do not necessarily produce better ontologies or plans.** Although one may expect the largest models to generate the most structured plans, but NextCoder-32B performance is surpassed by significantly smaller Qwen models. Qwen-32B yields the highest performance (0.43), and even Qwen-7B outperforms Gemma-12B, indicating that model family and training specialization matter more than raw parameter count.

**(2) Qwen models produce more stable and semantically consistent plan structures.** Models from the Qwen family demonstrate higher robustness in generating hierarchical task decompositions that align with our ontology constraints. This leads to more reliable curriculum construction and more effective RL training.

**(3) Some widely used LLMs fail to benefit from the alignment stage.** We also conducted experiments with several other well-known models, including *microsoft/phi-4*, *mistralai/Mistral-7B-Instruct-v0.2*, and *openai/gpt-oss-20b*, and found that the alignment stage does not provide any measurable benefit for them. Despite explicit prompt constraints on which subtasks should be used, these models tend to generate large numbers of synonymously similar subtasks. Consequently, the set of goals that the agent must recover becomes even larger than when instructions are provided directly, rendering it impractical to run the full pipeline with these models.

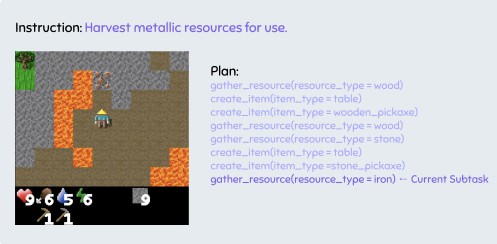

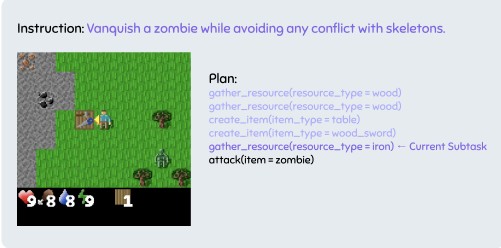

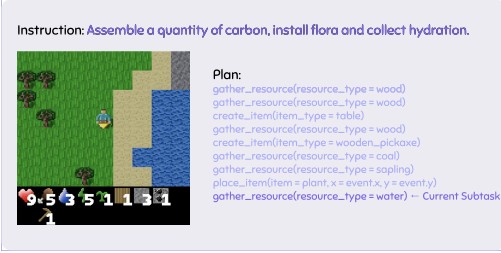

Figure 6: Example of instructions and corresponding plans

# D    CRAFTEXT

To provide a concrete example of our method,
Figure 6 visualizes the agent's state at a single timestep. The CrafText environment, shown on the left, is a dynamic grid-world where the agent must gather resources, craft items, and navigate diverse terrains to survive and complete tasks.

The core of our approach lies in the hierarchical decomposition of complex goals. As shown on the right, a high-level instruction, which may be ambiguous or require long-term planning (e.g., "Craft an iron pickaxe."), is first translated into a deterministic, multi-step plan. Each step in this plan constitutes a distinct subtask.

Crucially, the agent's policy is not conditioned on the entire plan. Instead, it focuses solely on the currently active subtask. This transforms a challenging long-horizon problem into a more tractable sequence of short-horizon tasks. The agent's objective at any moment is to complete the highlighted subtask and then invoke the DONE action. For example, optimal agent can choose DONE action based on the inventory state (when completing subtasks such as collecting resources and crafting items), player status (for subtasks that are related to eating, drinking or sleeping) or map state (for subtasks such as placing blocks).

Upon successful completion, the framework provides the next subtask in the sequence, guiding the agent through the overall plan until the final goal is achieved.

For our work we used a variation of Easy Craftext dataset EASY-STRICT, which introduces more strict instruction completition protocol. The structure of the dataset is as follows:

Training Set:

- Atomic: Single, indivisible goals (e.g., "Craft a furnace").
- Combo: Sequences of multiple atomic goals (e.g., "Craft furnace and then collect wood").
- Crucially, each instruction in the training set also has a paraphrased version to encourage linguistic robustness from the start.

Test Sets (Out-of-Distribution):

- Paraphrases: Contains the same underlying goals as the Combo training set, but expressed with novel vocabulary and syntax. This tests robustness to linguistic variation.
    - Training Combo: "Consume beef and create a stone pickaxe."
    - Test Paraphrase: "Eat steak and forge a stone pickaxe." or "Devour cow meat and create a stone pickaxe."
- New Objects: Introduces new combinations of atomic goals that appeared during training but never occurred together in a single instruction in the training set. This directly tests compositional generalization. These instructions also come with their own paraphrases.
- Training contained: "Consume beef" and "Forge a stone pickaxe" and "Forge a stone blade" as separate atomic or part of other combos.
- Test New Object: "Consume beef and forge a stone blade." or "Eat cow meat and create a sword from stone."

This structure allows us to rigorously dissect the agent's capabilities: learning from language (Atomic/Combo), generalizing to new phrasing (Paraphrases), and generalizing to new goal combinations (New Objects).

# E    COMPLETE SUPERIGOR TRAINING PIPELINE

The SuperIgor framework integrates multiple components that exchange specific inputs and outputs during training. Below we describe the key data flows between components:

**Component Interfaces:**

- **LLM Planner ($f_{\mathbf{LLM}}$)**

    - **Input:** Instruction $I$
    - **Output:** Candidate plan $\mathcal{P}$ consisting of a sequence of subtasks from subtask bank $\mathcal{B}$

- **RL Policy ($\pi_\theta$)**

    - **Input:** Environment observations $o_t$, current plan step DistilBERT [CLS] embedding $p_{\phi(t)}$ of a plan $\mathcal{P}$
    - **Output:** Action $a_t$ from extended action space containing default Craftext actions and additional DONE action that gives the agent the next plan step embedding $p_{\phi(t+1)}$ of a plan $\mathcal{P}$

The complete training procedure integrating all components is summarized in Algorithm 5

---

**Algorithm 5** Complete SuperIgor Training Pipeline

---

**Require:**
1: Environment $\mathcal{E}$
2: Instruction dataset $\mathcal{D}_{\text{train}} = \{I_1, I_2, \ldots, I_N\}$
3: Initial LLM planner $f_{\text{LLM}}$ with parameters $\theta_{\text{LLM}}$
4: Initial RL policy $\pi_\theta$ with parameters $\theta_{\text{RL}}$
5: Mastery threshold $\tau$, number of cycles $C$
**Ensure:**
6: Optimized planner $f_{\text{LLM}}^*$
7: Trained policy $\pi_\theta^*$
8:
9: Initialize subtask bank $\mathcal{B} \leftarrow \emptyset$
10: Initialize candidate plans $\mathcal{P} \leftarrow \{\}$
11: Initialize mastered subtasks $\mathcal{M} \leftarrow \emptyset$
12:
13: **Initial Plan Generation (Cycle 1):**
14: Extract subtasks: $\mathcal{S} \leftarrow f_{\text{LLM}}(\mathcal{D}_{\text{train}})$
15: Build ontology: $\mathcal{O} \leftarrow \text{BuildOntology}(\mathcal{S}, f_{\text{LLM}})$
16: Generate initial plans: $\mathcal{P}_{\text{initial}} \leftarrow \text{ExpandPlans}(\mathcal{D}_{\text{train}}, \mathcal{O})$
17:
18: Fine-tune $f_{\text{LLM}}$ on $\mathcal{P}_{\text{initial}}$ using SFT
19: Generate training plans: $\mathcal{P} \leftarrow f_{\text{LLM}}(\mathcal{D}_{\text{train}})$
20:
21: **for** cycle $c = 1$ **to** $C$ **do**
22:
23:     **Policy Training with Skill Curriculum:**
24:     Train $\pi_\theta$ on $\mathcal{P}$ using PPO with Skill Curriculum Learning
25:     Update mastered subtasks $\mathcal{M}$ based on success rates
26:
27:     **Plan Validation:**
28:     Execute $\pi_\theta$ with plans $P$ for multiple seeds
29:     For every plan $P$ in compute average success rate $SR(p)$
30:     Construct preference dataset $D_{\text{pref}}$
31:
32:     **LLM Fine-tuning:**
33:     Fine-tune $f_{\text{LLM}}$ on $\mathcal{D}_{\text{pref}}$ using DPO
34:
35:     **Plan Generation:**
36:     Select plans for new training epoch: $\mathcal{P} \leftarrow \text{SelectPlans}(f_{\text{LLM}}, \mathcal{D}_{\text{train}}, \mathcal{P})$
37: **end for**
38:
39: **return** $f_{\text{LLM}}, \pi_\theta$

---

## F DPO PLANS REPRIEORETIZATION

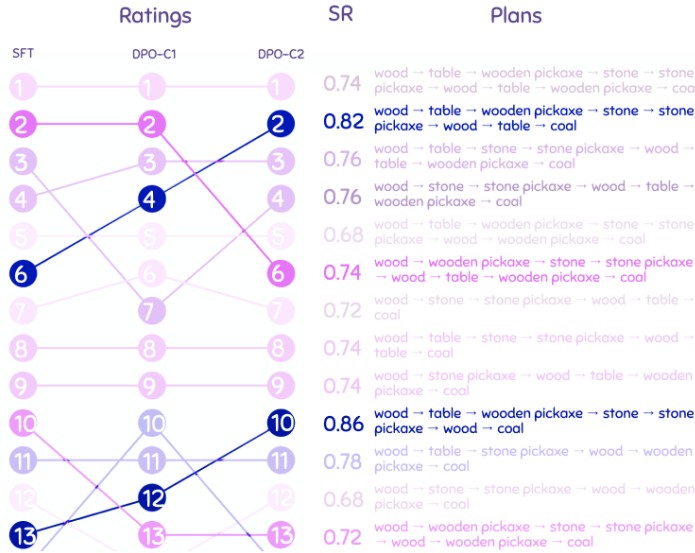

Figure 7: Example of DPO plan reprioritization for the instruction:*"Forge a stone pickaxe and mine coal"*

## G TRAINING DETAILS: POLICY OPTIMIZATION

Our low-level policy, which is responsible for executing individual subtasks, is trained using Proximal Policy Optimization (PPO). The agent's goal at this stage is to learn an optimal strategy for completing a given subtask based on its visual observations. The standard clipped surrogate objective for PPO is defined as:

$$\mathcal{L}^{\text{PPO}}(\theta) = \mathbb{E}_t \left[ \min \left( \rho_t(\theta) \hat{A}_t, \, \text{clip}(\rho_t(\theta), 1 - \epsilon, 1 + \epsilon) \hat{A}_t \right) \right],$$

where $\rho_t(\theta) = \frac{\pi_\theta(a_t|o_t)}{\pi_{\theta_{\text{old}}}(a_t|o_t)}$ is the probability ratio and $\hat{A}_t$ is the estimated advantage at timestep $t$.

Our agent's policy and value functions are parameterized by a single neural network with a shared multimodal feature extractor and separate actor and critic heads. The visual stream processes the $63 \times 63$ pixel image with 3 channels observations using a three-layer Convolutional Neural Network (CNN). Each convolutional layer utilizes 32 filters with a $5 \times 5$ kernel, followed by a ReLU activation and max-pooling. For the language stream, textual instructions are encoded using a pre-trained BERT model (`bert-base-uncased`), and we use the embedding of the `[CLS]` token as the final text representation.

The flattened output of the CNN and the text embedding are then concatenated to form a unified multimodal representation. This combined feature vector is fed into two separate feed-forward networks: the **actor head**, which outputs the logits for the categorical action distribution, and the **critic head**, which outputs a scalar estimate of the state-value function.

## H LLM FOR PLANNING IN INSTRUCTION FOLLOWING TASK

We conducted an additional comparison with prior works dedicated to solving the task of following language instructions by incorporating planning with language models, in order to illustrate the applicability of our approach and how it differs from existing methods. We examined whether current approaches can be used without predefined skills and verification functions, whether there

exist frameworks for training a low-level strategy and a planner, and we were also interested in the size of the model used for planning. The results of this comparison are presented in the table 3.

Table 3: Comparison of LLM-based planning methods across subgoal extraction, RL usage, reward specification, and model size.

| Method | Link | Operates without predefined skills | LLM Planer Training | Low-Level Policy Training | Work with Sparse Reward | Plan Model Size $< 20B$ | Plan Model Name |
|---|---|---|---|---|---|---|---|
| Plan-Seq-Learn (PSL) | link | ✗ | ✗ | ✓ | ✗ | ✗ | GPT-4 |
| DEPS | link | ✗ | ✗ | ✗ | N/A | ✗ | ChatGPT |
| SayCan | link | ✗ | ✗ | ✓ | ✗ | ✗ | PaLM 540B |
| Translated LLM | link | ✗ | ✗ | ✗ | N/A | ✓ | GPT-3 and Codex-12B |
| Few-shot Subgoal Planning with LMs | link | ✗ | ✗ | ✗ | N/A | ✓ | GPT-2-XL |
| IGOR | link | ✗ | ✓ | ✓ | ✗ | ✓ | Gemma-7B |
| **SuperIgor (ours)** | ... | ✓ | ✓ | ✓ | ✓ | ✓ | Qwen1.5-14B |

## I  TRAINING DETAILS: LLM FINE-TUNING

To improve the high-level planner (the LLM), we employ a reinforcement learning-based feedback loop. The planner generates a sequence of subtasks (a plan), which is then executed by the PPO agent. The final outcome of the agent's execution (e.g., task success or failure, efficiency) serves as a signal to update the planner.

**Direct Preference Optimization (DPO).** This method aligns the model toward preferred completions using pairwise preference data. The DPO loss is:

$$\mathcal{L}^{\text{DPO}} = -\log \sigma \left( \beta \left( \log \pi(x^+ \mid q) - \log \pi(x^- \mid q) \right) \right),$$

where $x^+$ and $x^-$ are preferred and less preferred plans for instruction $q$, and $\beta$ is a temperature parameter.

## J  COMPUTE RESOURCES

All experiments were conducted on a high-performance computing cluster equipped with nodes containing 1 NVIDIA A100 GPU with 80 GB of VRAM. Each node was powered by an 12 CPU Cores CPU with 96 GB of system RAM.

The total computational budget can be broken down into two primary stages:

**Policy Training and Evaluation.** The primary computational cost stems from training the PPO agent. Each full training run for a single configuration up to 10 billion environment steps took approximately 120-150 GPU-hours. Reproducing all presented experiments, including the baseline comparisons and ablation studies, required a total of 10 such training runs.

**LLM Traininga and Generation.** The initial generation of plans using the Qwen2.5-14B-Instruct model for the entire dataset required approximately X GPU-hours on a single NVIDIA A100 GPU. Epoch of finetuning LLM with DPO on evaluated plans takes approximately 15 GPU-hours.

In total, we estimate the full computational cost to reproduce all results presented in this paper to be approximately 2000-2500 GPU-hours.

## K  LIMITATION

Although our proposed method demonstrates a promising direction for integrating large language models with reinforcement learning for instruction-following tasks, it is not without limitations.

A primary limitation is the ambiguity in the attribution of failures. When the RL agent fails to complete a given plan, it is difficult to determine whether the failure stems from a flawed plan generated by the LLM or from inadequately trained policy in the RL agent. This ambiguity complicates the fine-tuning process for the language model, as the feedback signal may incorrectly penalize a viable plan that the agent was simply unable to execute. This can lead to a noisy training signal and potentially degrade the LLM's planning capabilities.

## L  AGENT'S PLAN FOLLOWING

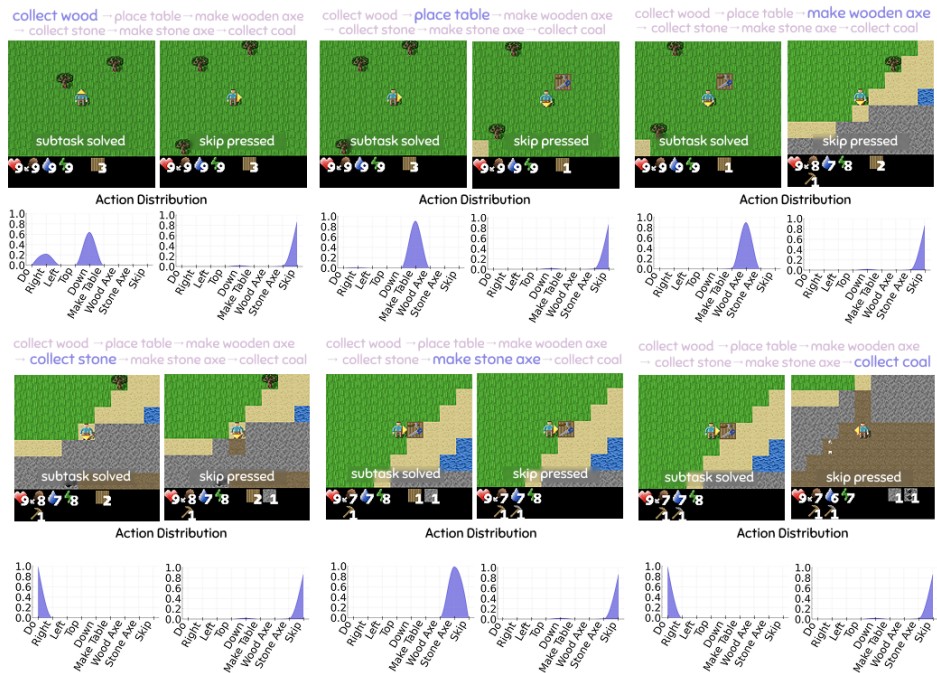

Figure 8: Example of how the agent follows the plan and chooses actions. For each subtask, there are two frames: the first shows the observation up to the moment when the agent takes the action that completes the subtask, along with the action distribution at that time; the second shows a few timesteps later, when the agent decides to skip the subtask in order to solve a new one.

## M  LLM USAGE

Large language models were employed solely for the purpose of refining and revising textual content, focusing on aspects such as grammar, spelling, and word selection.

## N  TRAINING DETAILS: HYPERPARAMETERS

The hyperparameters for our experiments are detailed in Table 4. For the PPO agent training, we adopt the configuration from the original CrafText baseline study (Volovikova et al., 2025). For the LLM planner fine-tuning, we use a Q-LoRA approach with a comprehensive set of parameters optimized for efficient large model training.

Table 4: Hyperparameters used for training the low-level agent and fine-tuning the high-level planner.

| Hyperparameter | Value |
| --- | --- |
| *PPO Agent Training* | |
| Learning rate | 0.0002 |
| Discount factor ($\gamma$) | 0.99 |
| GAE lambda ($\lambda$) | 0.95 |
| Clipping epsilon ($\epsilon$) | 0.2 |
| PPO epochs | 4 |
| Number of minibatches | 8 |
| Entropy coefficient | 0.01 |
| Value function coef. | 0.5 |
| Activation function | Tanh |
| Hidden layer size | 512 |
| *LLM Planner Fine-Tuning* | |
| Base model | Qwen2.5-14B-Instruct |
| Training epochs | 1 |
| Learning rate (SFT) | 2e-4 |
| Learning rate (DPO) | 1e-5 |
| Beta (DPO) | 0.5 |
| Optimizer | Paged AdamW (32-bit) |
| LR scheduler | Cosine |
| Warmup ratio | 0.03 |
| Batch size (per device) | 16 |
| Gradient accumulation | 1 |
| Gradient clipping norm | 0.3 |
| Weight decay | 0.001 |
| Mixed precision | bf16 |
| *LoRA Configuration* | |
| LoRA rank (r) | 64 |
| LoRA alpha ($\alpha$) | 16 |
| LoRA dropout | 0.1 |
| *Quantization (4-bit)* | |
| Quantization type | nf4 |
| Compute dtype | float16 |

