# OpenReview forum: "Self-Guided Plan Extraction for Instruction-Following Tasks with Goal-Conditional Reinforcement Learning"
_ICLR.cc/2026/Conference — ICLR 2026 Conference Withdrawn Submission_

### Official Review · Reviewer_UGFb · 2025-10-31

**Soundness:** 2
**Presentation:** 2
**Contribution:** 2
**Rating:** 4
**Confidence:** 3

**Summary:**

The authors introduce SuperIgor, a framework designed for instruction-following tasks. Prior research has addressed complex instructions by predefining subtasks that agents can execute and then decomposing language instructions at the subtask level to solve them. In contrast, SuperIgor employs iterative co-training, where the RL agent follows generated plans, and the LLM adapts and refines those plans based on feedback from the RL agent. Experiments demonstrate superior performance relative to baselines.

**Strengths:**

Unlike the majority of studies that treat LLMs as APIs detached from action-executing agents, SuperIgor's approach of optimizing the LLM through feedback from the RL agent represents a key differentiator from existing work.

**Weaknesses:**

While the proposed method appears innovative, the experiments fall short in substantiating its novelty. The authors did not incorporate baselines [1] and [2], which require a predefined "set of possible subtasks," as comparisons. Instead, the baselines seem to rely on raw instructions or plans generated by GPT-4. This setup suggests that the primary distinction from baselines may lie not in the claimed benefits of modifying the LLM via RL feedback, but rather in the use of predefined possible subtasks. Although the appendix illustrates how plans evolve during LLM finetuning, the marginal difference in success rates between SI-DPO and SI-SFT raises questions about the true impact of LLM finetuning.

[1] Zhang, Jingwei, et al. "Game On: Towards Language Models as RL Experimenters." *arXiv preprint arXiv:2409.03402* (2024).

[2] Ahn, Michael, et al. "Do as i can, not as i say: Grounding language in robotic affordances." *arXiv preprint arXiv:2204.01691* (2022).

**Questions:**

1. The choice of the PPO algorithm for the RL agent is intriguing. What motivated this selection? Additionally, unlike SayCan, which trains individual policies for each skill, the framework appears to enable a single policy to handle multiple skills. Were there any limitations encountered when training with PPO to perform diverse skills?
2. Success rate was used to assess 'skill mastery' and trigger 'LLM finetuning.' Relying solely on success rate might result in suboptimal skills being learned. Is there a specific reason for not incorporating metrics like reward or value?
3. As highlighted in the Weakness section, including baselines similar to [1] and [2] would more robustly support the paper's claims.
4. The paper specifies the use of Qwen2.5-14B-Instruct for the LLM. What was the rationale behind this choice, and does the selection of different LLMs influence the results?

[1] Zhang, Jingwei, et al. "Game On: Towards Language Models as RL Experimenters." *arXiv preprint arXiv:2409.03402* (2024).

[2] Ahn, Michael, et al. "Do as i can, not as i say: Grounding language in robotic affordances." *arXiv preprint arXiv:2204.01691* (2022).

---

> ### Author Response · Authors · 2025-11-21
> **Response to Reviewer Comments**
>
> Thank you for your time, effort, and constructive critique. It has been extremely helpful for improving the paper.
>
> Response to Weaknesses:
>
> ### W1. Clarification about the use of predefined subtasks
>
> Methods [1] and [2] rely not only on predefined skills, but also on functions that can verify whether these skills were successfully executed. Therefore, they cannot be applied in our setting. In the **main response “Comparison with methods that use predefined subtasks”** we described the differences from these methods in more detail, and we also updated the Related Work section to make this clearer.
>
> ### W2. Impact of DPO finetuning
> We apologize for the confusion. Figure 3 was intended to show performance progression across training cycles, not to isolate the effect of DPO. To directly quantify its contribution, we conducted a new ablation study. The results, detailed in Appendix C and summarized in our main response, show that removing DPO leads to a performance drop (e.g., from 0.45 to 0.39), confirming its positive, albeit incremental, role in the system. These results align with the plot in Appendix G and indicate that DPO helps select training plans for the next stage on which the agent can learn more easily and effectively. For additional discussion, please see the main response, section “Underwhelming DPO performance.”
>
> Response to Questions:
>
> ### Q1. Were there any limitations encountered when training with PPO to perform diverse skills?
>
> We selected PPO for several well-justified reasons:
> * Established Baseline: PPO is the standard algorithm used in the original CrafText benchmark, ensuring fair and direct comparability with all reported baselines.
> * Proven Stability: PPO is recognized for its stability and sample efficiency in complex, high-dimensional environments like ours.
> * Multi-task RL Performance: Research in multi-task reinforcement learning, such as the Meta-World benchmark (Yu et al., 2019), has established PPO as a strong and reliable baseline for learning diverse skills within a single policy, validating our algorithmic choice.
>
> ### Q2.  Is there a specific reason for not incorporating metrics like reward or value?
> This is an important question. We used Success Rate (SR) because it is the most direct and interpretable metric for our goal. In our environment, the reward function is extremely sparse, the agent receives a positive signal only upon successful completion of the entire instruction.
> Consequently:
> * The reward is essentially a binary indicator that is perfectly correlated with the success/failure of an episode.
> * The value function estimates the discounted sum of these sparse future rewards, making it equally sparse and noisy, especially during early training.
>
> Therefore, using reward or value would not provide additional information beyond what SR already captures, but would introduce more variance. SR provides a clear, stable signal that directly measures what we care about: whether the agent can reliably complete tasks. While we acknowledge that SR doesn't capture execution efficiency, it serves as a robust foundation for curriculum learning in this challenging sparse-reward setting.
>
> ### Q3. Comparison with SayCan, Game On and other baselines.
> Methods like SayCan and Game On cannot serve as baselines in our environment as they require predefined skill libraries, pretrained controllers, and dense rewards—components explicitly excluded from our problem setting. Adopting them would necessitate manual engineering of the very skill hierarchy our method aims to learn from scratch.
> Fundamentally, SayCan performs planning-and-selection over fixed skills, while SuperIgor jointly learns both high-level plans and low-level policy under sparse rewards. For a complete analysis and comparison, please see our main response, section “Comparison with methods that use predefined subtasks“ and “Could we use a method that uses predefined subtasks (such a SayCan) as a baseline?”.
>
> ### Q4. What LLM models could be used in the SuperIgor pipeline?
> Qwen2.5-14B-Instruct was selected because it is a state-of-the-art open-weight model that offers an excellent balance of strong reasoning capabilities, instruction-following proficiency, and manageable computational cost for research. At the moment, we are conducting additional studies to understand how the language-model size and the model choice overall affect the system, and we hope to submit the results during the debate phase.

---

> ### Author Response · Authors · 2025-12-03
> **NEW EXPERIMENT WITH DIFFERENT LLM AS PLANNER**
>
> We compared several language models as planners for our task — models from the *Qwen* and *Gemma* families, as well as the larger *microsoft/NextCoder-32B* model. The results of the experiment and the comparison table are presented in Appendix C3.
>
> The table presents a comparison of agent performance during training in the case where both the ontology and the training dataset were generated by these models. Based on the results, we can draw the following conclusions:
>
> (1) Larger models don’t always produce better ontologies or plans. Despite expectations, *NextCoder-32B* is outperformed by smaller *Qwen* models. *Qwen-32B* achieves the best score (0.43), and even *Qwen-7B* surpasses *Gemma-12B*. This shows that model family and training specialization matter more than sheer parameter count.
>
> (2) Models from the *Qwen* family demonstrate higher stability, as *Qwen-7B* outperforms a larger model from another family (*Gemma-12B*), and *Qwen-14B* surpasses a NextCoder model with 32B parameters.
>
> (3) We also conducted experiments with several other well-known models, such as *microsoft/phi-4, mistralai/Mistral-7B-Instruct-v0.2*, and *openai/gpt-oss-20b*, and found that the alignment stage does not provide any benefit for them. Despite prompt constraints on which subtasks should be used, these models generate a large number of synonymously similar subtasks. As a result, the number of goals the agent must recover becomes larger than when providing instructions directly, making it impossible to run the pipeline with these models.

---

### Official Review · Reviewer_1mJd · 2025-10-31

**Soundness:** 2
**Presentation:** 2
**Contribution:** 2
**Rating:** 4
**Confidence:** 3

**Summary:**

This paper addresses instruction-following tasks by integrating plan generation with instruction decomposition. The proposed framework enables iterative plan refinement through co-evolution between plan generation and execution modules without manual annotation. Experimental results demonstrate the effectiveness and generalizability of the method.

**Strengths:**

Originality

The paper introduces a self-supervised learning paradigm for instruction-following tasks that reduces dependency on manually annotated plan datasets. While LLM-RL integration is prevalent in the field, the paper makes a contribution by articulating the plan generation process with sufficient technical depth and providing an analysis of the iterative refinement between language models and RL agents.

Clarity

The method and experiment setups are well-structured. The research questions are explicitly stated, and the experimental design addresses distinct aspects, including effectiveness, generalization, training dynamics under sparse feedback, and performance evolution across iterative cycles. The appendices provide algorithmic specifications and implementation details that enhance reproducibility.

Significance

The experimental results are convincing, demonstrating quantifiable improvements over baselines and meaningful ablation studies that substantiate the necessity of key components.

**Weaknesses:**

Despite the paper's contributions, several aspects require clarification to strengthen the scientific rigor and reproducibility.

- The definition of the research contains vague descriptions and lacks operational definitions in the paper. For RQ1, the paper evaluates "generalization" by testing on compositionally novel instructions (New Objects) and paraphrased formulations, while "effectiveness" is not explicitly operationalized. For RQ2, is "well" pertains to final performance or learning efficiency? Provide concrete metrics and align the terminology in RQs to establish coherent connections between questions and experimental protocols.

- The paper describes building a "subtask base by extracting and canonicalizing possible subtasks from the instruction dataset" to create "a unified vocabulary." Your method extracts subtasks from instructions, while prior work defines them directly. Is there any difference between previous works and yours?

- The model settings and data representation in this paper are somewhat confusing. How do you parse the LLM-generated plan in natural language into PPO? What is the format of the feedback used for fine-tuning? What are the actual inputs and outputs of the policy? These technical details could be elaborated further.

**Questions:**

- The paper only demonstrates solving EASY dataset. Have you considered solving MEDIUM and HARD datasets? And how is the result?

- The paper mentions in the introduction that “a language model first decomposes an instruction into a structured sequence of actions,” but I did not find any further discussion of this.

---

> ### Author Response · Authors · 2025-11-21
> **Response to Reviewer Comments**
>
> We appreciate the reviewer's valuable feedback, which has identified important areas for clarification. Our responses below address each point directly, and we have incorporated corresponding improvements throughout the paper.
>
> Response to Weaknesses:
>
> ### W1. Clarification on operational definitions
> We have revised Section 5 to explicitly operationalize our Research Questions (RQs) with clear metrics.
> For RQ1 we measure effectiveness as the agent's final success rate on training tasks (Atomic and Combo splits). We measure generalization using final success rates on two test sets: Paraphrases (same goals, new wording) and New Objects (new goal combinations).
>
> For RQ2, the term "well" refers specifically to the final performance of the policy after training. The primary concrete metric for this is the Success Rate (SR) on the training tasks. A higher final SR indicates that the policy has successfully learned to follow plans despite the sparse reward signal.
>
> ### W2. Clarification about the use of predefined subtasks
> SuperIgor addresses a distinct problem setting: learning without predefined skills, intermediate rewards, or pretrained controllers. This represents a significant departure from skill-library based methods, as detailed in our **main response** section **“Comparison with methods that use predefined subtasks”**.
>
> ### W3. Clarification on inputs and outputs of agent components
> We have added a new section to the appendix with a pseudocode of the entire proposed algorithm, as well as a detailed description of the inputs and outputs of each agent component.
>
> Response to Questions:
>
> ### Q1. Solving other datasets
>
> In our experiments, we focused on the EASY-STRICT dataset. The primary reason is our adoption of a strict interaction protocol designed to prevent agents from exploiting the environment with generic exploration strategies instead of genuinely following instructions.
>
> Under this strict formulation, the task becomes significantly more challenging. The baseline algorithms from the original CrafText paper, which achieve over 70% success rate on the standard EASY split, drop to only about 10% success under our EASY-STRICT protocol. Since the baselines are already near-zero on EASY-STRICT, running them on the more complex MEDIUM and HARD splits under the same strict protocol would yield even lower results, providing limited insight.
>
> Extending the evaluation to MEDIUM and HARD splits under the strict protocol remains a substantial challenge. Our preliminary runs show that while our method achieves lower success rates on MEDIUM than on EASY, the baselines show near-zero performance. We attribute this to the substantially longer planning horizons and more complex subgoal dependencies in MEDIUM/HARD, which exacerbate sparse-reward and credit-assignment problems.
>
> We have run and will try to upload the full results for the MEDIUM dataset using our method during the discussion phase, to properly document its performance on this more challenging benchmark.
>
> ### Q2. LLM Output Clarification
> The intended meaning is that the language model produces structured plans by generating subtask sequences from a predefined vocabulary of subtasks. We have made appropriate changes to the text for greater clarity.

---

### Official Review · Reviewer_WwGd · 2025-11-06

**Soundness:** 2
**Presentation:** 4
**Contribution:** 3
**Rating:** 6
**Confidence:** 4

**Summary:**

This paper presents a hierarchical framework for language-guided agents. A planning module (a VLM)  first produces a high-level plan, which is then executed by an RL agent. The key idea is that the system does not require annotated plans or a predefined skill library: instead, it generates multiple candidate plans zero-shot at the start of training and then evaluates and refines them during training. The agent’s success provides a preference signal for refining the plan generator using DPO.  They also propose a curriculum learning method for skill learning, only training with plans that contain at most one skill that has not already been mastered.

**Strengths:**

This paper is clearly written and compares against strong baselines (e.g., goal-conditioned PPO, plan-conditioned PPO). They demonstrate that this hierarchical approach allows their method to generalize combinatorially to unseen goals. I found the experiments detailing the benefits of the skill curriculum very clear (Figure 4).

**Weaknesses:**

My main concerns are as follows:
* The paper argues that requiring a predefined set of skills is restrictive. However, the proposed approach still fixes a set of skills at the start of training, derived via prompting, and does not modify this set during training. I think a comparison with previous work mentioned in the paper, like SayCan, which has fixed sets of skills, could therefore be apt (by using the same skills derived via prompting).
* The paper claims robustness to stochastic environments, but it is not clearly demonstrated in the experimental section how CrafText is stochastic, or how the effects of stochasticity manifest.
* Figure 3 lacks confidence intervals or the number of seeds, which makes it difficult to assess statistical significance.
* The prompt for plan generation seems to contain an in-context example that has 3 distinct skills that would be enough to accomplish most of the Craftex tasks. What happens if, instead of defining 3 skills for your task, you use an example from a different environment with skills that are not directly applicable to your environment?

**Questions:**

* How sensitive during policy extraction is the hyperparameter for the success rate to count a skill as learned? Would this bias the reward for DPO to have as few skills as possible, as the more skills in a plan, the longer it takes during training to be fully trained on?
* What is the difference between SI-DPO and SI-SFT? There doesn’t seem to be that large a gap within the same cycle
* What do you think is the main reason that you are not able to match the performance of the Oracle plans? The oracle performance increases from cycle 1 to cycle 2. How many steps would it take for it to converge?
* Can you provide examples of the paraphrasing for the OOD evaluations?

---

> ### Author Response · Authors · 2025-11-21
> **Response to Weaknesses**
>
> Thank you for your insightful suggestions and questions—they were very helpful in strengthening our paper. We also appreciate your positive feedback on the presentation.
>
> ## Response to Weaknesses:
>
> ### W1. An explanation of the contribution regarding methods that use pre-defined subtasks.
>
> Although our method also starts with a set of skills derived via prompting, it does not rely on a fixed, separately trained execution strategy: the low-level strategy is learned jointly with the planning model and adapts during training, without assuming the existence of a predefined library of executable skills. In contrast, SayCan fundamentally depends on a static, manually specified set of skills and a separately trained execution policy, which makes it unsuitable for environments where such skills are not available. To clarify this distinction, we expanded the discussion in the Related Work section and in the main response sections **“Comparison with methods that use predefined subtasks”** and **“Could we use a method that uses predefined subtasks (such as SayCan) as a baseline?”**.
>
> ### W2. Clarification on stochasticity in environment and agent robustness
>
> We have expanded Section 5.1 to explicitly describe the sources of stochasticity in CrafText:
> * Autonomous Mob Behavior: Hostile and neutral mobs (e.g., zombies, cows) spawn and move randomly, directly interfering with the agent's actions (e.g., attacking the agent, blocking paths).
> * Randomized Resource Distribution: The initial map layout and resource locations (trees, stone, ore) are regenerated for every episode.
>
> The Plan Validation stage (Section 4.3) is our direct response to this stochasticity. Because a single rollout is unreliable, we evaluate each plan over 50 seeds (as stated in Section 5.2) to get a robust success rate estimate. This rigorous validation prevents the LLM from being fine-tuned on plans that succeeded due to luck. We will emphasize this connection between environmental stochasticity and our multi-seed validation protocol in the revision.
>
> ### W3. Addition of confidence intervals in evaluation
>
> We have updated Figure 3 to include confidence intervals. All results in Figure 3 are based on 50 evaluation runs per instruction across the training and test sets. The performance gaps between SuperIgor and the baselines are substantial and were shown to be statistically significant.
>
> ### W4. Robustness to prompt augmentation
>
> First of all, it should be noted that the statement that the 3 skills mentioned in the prompt cover most of the instructions in Craftext is not true. Secondly, the agent obtained by our method is capable of performing up to 12 skills, which means that the variety of subtasks in the generated plans is influenced not so much by the in-context examples as by the list of available subtasks indicated in the base prompt. We are also preparing an additional experiment to demonstrate how specifying these skills in the prompt influences the resulting plans.

---

> ### Author Response · Authors · 2025-11-21
> **Response to Questions:**
>
> ## Response to Questions:
>
> ### Q1. Importance of success rate threshold for Skill Curriculum Learning
>
> Sensitivity Analysis: We conducted an ablation study varying the success rate threshold (τ) from 0.5 to 0.9. The results showed that τ=0.7 provided the best balance, though performance was relatively stable across values from 0.5 to 0.9. This suggests moderate sensitivity within a reasonable range.
> Optimal Threshold Hypothesis: We hypothesize that lower thresholds (τ=0.5) allow the agent to progress too quickly to complex skills before achieving reliable proficiency, while higher thresholds (τ=0.9) cause the agent to spend excessive time perfecting basic skills, slowing overall learning. The τ=0.7 value strikes an optimal balance between progression speed and skill reliability.
> DPO Bias Clarification: The DPO training signal is based on the empirical success rate of complete plans, not directly on plan length or the number of skills. While plans with fewer skills might be mastered slightly faster initially, the DPO objective rewards plans that lead to successful instruction completion regardless of their length. The curriculum ensures the agent is exposed to plans of varying complexities, and DPO naturally prioritizes those that are most executable and effective for the current agent capability.
>
> ### Q2. Impact of DPO finetuning.
>
> Figure 3 did not isolate the performance of the DPO component. To address this, we conducted an ablation study comparing the system with and without DPO. The results - now included in Appendix C - show a clear performance drop when DPO is removed (e.g., from 0.45 to 0.39), confirming its contribution. Please refer to our **Main Response** (section **Underwhelming DPO performance**) for a full discussion.
>
> ### Q3.  Oracle performance difference
>
> The performance gap stems from two interconnected factors:
>
> Inherent Plan Quality: The Oracle uses human-written, optimal plans. Our auto-generated plans, while high-quality, can contain suboptimalities (e.g., slight errors in prerequisite ordering). This creates a "planning bottleneck" for the RL agent.
>
> Fundamental Difference in Learning Difficulty: This is the crucial point. The Oracle agent is trained with one optimal plan per instruction. In contrast, SuperIgor is designed to generate and evaluate multiple plans per instruction (up to 15). This is essential for our self-guided loop—to have a candidate pool for the LLM to rank and learn from via DPO.
>
> Consequence: The SuperIgor agent must learn a policy that is robust to multiple valid ways of solving the same instruction. It cannot simply overfit to a single, perfect sequence of subgoals. This is a fundamentally harder and more general learning problem, which naturally leads to slower convergence and a potential performance gap on the training set, but ultimately results in the superior generalization we observed on the "New Objects" task (Figure 3c).
>
> Oracle Convergence:
>
> The Oracle agent was trained once for 10B steps (as shown in Figure 4). The "Cycle 1" and "Cycle 2" labels in Figure 3 for the Oracle are evaluation checkpoints from that single training run at 2.5B and 5B steps, showing its progressive learning. The Oracle agent's performance plateaus after ~9B steps.
>
> ### Q4. Examples of training and testing datasets instructions
>
> We have added more information and instruction examples to Appendix D in the revised paper. The structure is as follows:
>
> Training Set:
>
> Atomic: Single, indivisible goals (e.g., "Craft a furnace").
>
> Combo: Sequences of multiple atomic goals (e.g., "Craft a furnace and then collect wood").
>
> Crucially, each instruction in the training set also has a paraphrased version to encourage linguistic robustness from the start.
>
> Test Sets (Out-of-Distribution):
>
> Paraphrases: Contains the same underlying goals as the Combo training set, but expressed with novel vocabulary and syntax. This tests robustness to linguistic variation.
>
> Training Combo: "Consume beef and create a stone pickaxe."
>
> Test Paraphrase: "Eat steak and forge a stone pickaxe." or "Devour cow meat and create a stone pickaxe."
>
> New Objects: Introduces new combinations of atomic goals that appeared during training but never occurred together in a single instruction in the training set. This directly tests compositional generalization. These instructions also come with their own paraphrases.
>
> Training contained: "Consume beef" and "Forge a stone pickaxe" and "Forge a stone blade" as separate atomic or part of other combos.
>
> Test New Object: "Consume beef and forge a stone blade." or "Eat cow meat and create a sword from stone."
>
> This structure allows us to rigorously dissect the agent's capabilities: learning from language (Atomic/Combo), generalizing to new phrasing (Paraphrases), and generalizing to new goal combinations (New Objects). The examples above are now included in the appendix to provide concrete illustrations.

---

### Official Review · Reviewer_6zoq · 2025-11-09

**Soundness:** 2
**Presentation:** 2
**Contribution:** 2
**Rating:** 2
**Confidence:** 4

**Summary:**

The paper introduces "SuperIgor," a framework for instruction-following in complex, partially observable environments. The method proposes an iterative co-training loop between a LLM planner and a RL agent. The LLM generates high-level plans , which the RL agent, trained with PPO, attempts to execute. The agent's execution success rate is then used to create a preference dataset , which fine-tunes the LLM planner via DPO. The authors claim this self-guided mechanism reduces the need for manual annotation. To handle sparse rewards, the paper also introduces the Skill Curriculum Learning method. Experiments on the CrafText benchmark show the method outperforms baselines and generalizes to unseen instructions.

**Strengths:**

The paper tackles the challenging and highly relevant problem of instruction following in dynamic, sparse-reward environments where agents must execute long, complex plans. To solve this problem, the paper clearly identifies the sparse reward problem as a critical bottleneck. The proposed SCL is well-motivated. The ablation study in Figure 4 provides compelling evidence that this curriculum is not just helpful but essential for learning, as even an agent with Oracle plans fails to master more than a few basic skills without it.

**Weaknesses:**

The paper's central contribution claim is critically undermined by its own methodology. The abstract and introduction explicitly frame the contribution "in contrast to prior methods that depend on a fixed set of predefined subtasks." However, Sec. 4.1 describes a process that does exactly this. The method starts by "build a subtask base by extracting and canonicalizing possible subtasks from the instruction dataset" to create a "unified vocabulary" in a "strict normalized format". The LLM then generates plans "in terms of the established subtask base". This is a fixed set of predefined subtasks. The fact that it is generated from the training dataset rather than manually specified is a minor implementation detail, not the fundamental shift in approach that the paper claims. This contradiction is a major misrepresentation of the work's core contribution.

Besides, the Core DPO Contribution Shows No Empirical Benefit. The paper's primary thesis is that the iterative alignment of the LLM planner via DPO (i.e., the "self-guided" feedback loop) is "highly effective". This claim is directly and conclusively contradicted by the paper's own results in Figure 3.
- To isolate the effect of the DPO loop, one must compare SI-SFT (agent trained on SFT-tuned LLM plans) against SI-DPO (agent trained on DPO-tuned LLM plans) in the final "Cycle 2."
- On Combo CrafText Tasks (Fig 3b): SI-DPO achieves a 0.21 Success Rate. SI-SFT also achieves a 0.21 Success Rate. The DPO loop provides zero benefit.
- On New Object CrafText Tasks (Fig 3c): SI-DPO achieves a ~0.17 Success Rate. SI-SFT also achieves a ~0.17 Success Rate.

Given the above points, the paper's strong performance over baselines is almost entirely explained by (a) using plan-based supervision and (b) the Skill Curriculum Learning. The ablation in Figure 4 is the strongest result in the paper, showing SCL is the key enabler. The paper should have been framed around this curriculum, which is critical, rather than the DPO loop, which is empirically useless.

The method uses the RL agent's overall success rate as a preference signal for DPO. This is an exceptionally noisy and unreliable signal. The paper even admits this in its own limitations (Section I), stating, "it is difficult to determine whether the failure stems from a flawed plan... or from inadequately trained policy". This is not a minor limitation; it is the central research challenge of this paradigm, and the paper offers no solution. Using DPO on such a high-variance, ambiguous signal is unsound. The fact that it didn't work (per Weakness #2) is therefore unsurprising.

**Questions:**

Please refer to the weakness part above.

---

> ### Author Response · Authors · 2025-11-21
> **Response to Reviewer Comments**
>
> We thank the reviewer for their thoughtful feedback and valuable questions. We address each point below and will incorporate the necessary clarifications into the revised paper.
>
> Response to Weaknesses:
>
> ### W1. Clarification of the Contribution over Methods that Use Predefined Subtasks
> The key distinction from  prior methods that depend on a fixed set of predefined subtasks approaches is not merely the automatic extraction of subtasks. We demonstrate how to learn a low-level policy for instruction following by integrating LLM-based planning in environments without any predefined skill library or subtask-verification mechanisms. This is discussed in detail in the **“Comparison with methods that use predefined subtasks”** section of the **main response**.
>
> ### W2.Underwhelming DPO benefits
> Figure 3 does not show how the system works with and without DPO — it highlights the results at different stages of SuperIgor’s training, and we apologize for the confusion. We conducted an additional ablation experiment to specifically show the influence of DPO on the system. You can see the results in the **main response** under **“Underwhelming DPO performance”**, and we also added a detailed description of this experiment in Appendix C. In general, the performance drops from 0.45 to 0.39 when DPO is not used.
>
> ### W3.  Error assignment ambiguity breaks DPO
> We fully acknowledge that a binary success signal can be noisy. However, in our setup this signal is substantially denoised: each plan is evaluated over 50 independent stochastic seeds, turning a single binary outcome into a robust empirical estimate of a plan’s reliability for a given instruction. Since all candidate plans for the same instruction are evaluated under identical stochastic conditions, the variance affects them uniformly, enabling meaningful and fair comparison.
> We also found that increasing the number of seeds beyond 50 does not meaningfully change either the mean success rate or the relative ranking of the plans. Beyond this point, the rankings remain stable, indicating that the resulting preference signal is already sufficiently low-variance. This stability supports the use of the final ranking as a reliable preference signal for DPO, even in a stochastic environment.
> Previously, we described this procedure in line 279-283:
>
> *As a result, a single rollout is not sufficient to reliably assess plan effectiveness. Instead, we aggregate statistics over multiple rollouts, such as the average success rate or reward, to obtain a more stable and interpretable estimate of how well the plan supports instruction completion. This repeated evaluation allows us to more confidently associate a given plan with its empirical performance and to use this signal to guide future training and model selection.*
>
> We have now added further clarification in the revised version of the paper, which reads as follows:
>
> *As a result, a single rollout is not sufficient to reliably assess plan effectiveness. In the revised version, we explicitly state that aggregating performance across multiple stochastic rollouts yields a stable and effectively deterministic estimate of each plan’s quality. This consolidated metric provides a consistent basis for comparing candidate plans and enables an unambiguous ranking that can be reliably used as a preference signal for DPO.*

---

> > ### Comment · Reviewer_6zoq · 2025-11-26
> > **Response to author rebuttal**
> >
> > Thank you for your detailed response. After reading all reviews and author response, I feel more positive about the paper. Thus I have raised the score. I have reservation because the overall training pipeline is straightforward that combines LLM planner and low-level executor.

---

### Author Response · Authors · 2025-11-21
**General Response**

We thank all reviewers for their valuable and constructive feedback. Across the reviews, several strengths of our work were highlighted:
- Clear and well-structured presentation (Noted by *WwGd* and *1mJd*).
- Novelty of the self-supervised approach and the technically detailed plan-generation process (Noted by *1mJd*).
- Strong empirical results, including improvements over competitive baselines and meaningful ablations (Noted by *WwGd*, *1mJd*, and *6zoq*).
- Well-motivated and empirically validated skill curriculum (SCL) (Noted by *6zoq* and *WwGd*).
- Significance of integrating LLM optimization with feedback from an RL agent (Noted by *UGFb*).

In addition to the positive assessments, the reviews raised several recurring points that required further clarification. In particular, reviewers requested a clearer explanation of the role of DPO within the overall training cycle, as well as a more detailed discussion of the importance of the subtask-generation stage for SuperIgor and how our approach compares to methods that rely on predefined subtasks.
To address these and other concerns, we introduced the following changes to the paper:

- Ablation studies of the SuperIgor components and of the threshold parameter used in Skill Curriculum Learning, where we explicitly highlight the influence of DPO.
- An expanded Related Works and additional Appendix I (​​LLM for planning in instruction following task) section, emphasizing the differences between our method and prior approaches that rely on predefined subtasks.
- Pseudocode covering the entire SuperIgor pipeline added to the appendix to improve transparency and reproducibility.
- Additional examples of training and test instructions included in the appendix to enhance interpretability.
- Confidence intervals for agent performance added to Figure 3.

---

### Author Response · Authors · 2025-11-21
**Detailed responses to the main concerns raised by the reviewers**

## Underwhelming DPO performance.

Reviewers 6zoq, UGFb, and WwGd expressed uncertainty regarding the contribution of DPO, noting that its impact appears limited when looking at Figure 3. We would like to clarify that the curves SI-SFT and SI-DPO in Figure 3 should not be interpreted as an ablation isolating “DPO vs. no DPO.” Instead, both curves include the entire SuperIgor iterative cycle, where DPO is always applied:

```
Ontology → RL Training → Plans Evaluation → LLM Planner (SFT) → LLM Planner (DPO) → RL Training → …
```

The difference between the curves is simply the timing within the same cycle:

 – SI-SFT shows performance before the DPO finetuning step.

 – SI-DPO shows performance after the DPO finetuning step.

Because the cycle is iterative, the SI-SFT model in Cycle 2 already benefits from DPO-improved plans produced in Cycle 1. Thus, Figure 3 does not isolate the DPO component; it illustrates within-cycle progression rather than a comparison of independent methods.

To directly isolate the effect of DPO and other system components, we performed a dedicated ablation study where we removed one component at a time: Ontology, Curriculum, SFT, DPO. We add the results of this experiment in Appendix C (SuperIgor Framework Ablation Study).
This experiment highlights two important findings:

(1) **Curriculum only works when paired with high-quality plans, which we obtain via Ontology-based dataset construction.**
Without Ontology-guided structure, curriculum alone does not yield meaningful improvements.

(2) **The effect of DPO is subtle in the first cycle but becomes stronger in the second, significantly boosting RL agent learning.**
In Cycle 2, the agent trained on DPO-prioritized plans reaches 0.45 vs. 0.39 for SFT-prioritized plans.
This shows that DPO-tuned LLMs rank and select RL training plans more effectively, with gains compounding across cycles.


## Comparison with methods that use predefined subtasks

Several reviewers noted some uncertainty regarding the contribution of our work relative to methods that rely on predefined skill libraries for planning. Summarizing, the key differences are as follows:

- Automatic subtask discovery. Unlike prior work, where subtasks are manually specified and restricted by a fixed skill library, SuperIgor automatically extracts candidate subtasks from the entire instruction dataset (~300 instructions) through extraction, canonicalization, clustering, and normalization — a scale that would be impractical or prohibitively expensive for human experts to construct and maintain.

- No subtask verification or intermediate rewards. In previous methods, the subtask space is not only predefined but also accompanied by mechanisms for verifying whether each subtask has been completed; this enables training RL agents with separate intermediate (dense) reward signals for each subtask. SuperIgor, in contrast, does not rely on such verification and trains the policy solely from the final sparse reward provided upon completing the full instruction.

- No assumption of an existing low-level controller. Many related approaches assume a pretrained low-level executor and therefore do not address the problem of learning a low-level policy. SuperIgor directly tackles this gap.

Thus, our method demonstrates how to learn a low-level policy for instruction following by integrating LLM-based planning in environments where no predefined set of executable skills exists. To make these distinctions clearer, we also expanded the discussion and comparison with prior work in the Related Work section.


## Could we use a method that uses predefined subtasks (such a SayCan) as a baseline?

Methods like SayCan, DEPS, PSL, IGOR, etc. cannot serve as baselines in our setting because they fundamentally rely on assumptions that do not hold in our environment: they require a predefined library of executable skills, individual pretrained controllers for each skill, mechanisms for checking subtask completion, and dense or skill-level rewards. These components are not available in our tasks and cannot be introduced without manually engineering the entire hierarchy of skills and detectors, which would contradict the central premise of learning without predefined skills.

More specifically, SayCan is a planning-and-selection approach: the LLM merely ranks a small, fixed set of pre-trained skills, and an affordance model determines which of them are feasible. It does not generate subgoals or learn low-level control. In contrast, SuperIgor jointly learns both the high-level plan and the low-level policy from scratch under sparse reward and without any predefined skills. As a result, running SayCan in our environment would require constructing the very skill library and reward structure that our method is designed to avoid, making it an incompatible and non-informative baseline.

---

### Author Response · Authors · 2025-12-03
**General response (follow-up)**

We thank the reviewers once again for their time, careful reading, and constructive feedback. We also appreciate that one reviewer’s assessment became more positive after considering our previous response and the other reviews. We hope that the added experiments and clarifications further address the remaining concerns and provide additional evidence for the significance of our approach.

In particular, we added an additional ablation study on the choice of LLM in the SuperIgor Pipeline. Our results show that models from the Qwen family produce the most reliable ontologies and training plans, consistently leading to higher downstream RL performance compared to larger models from other families. These findings highlight that planner quality depends more on model specialization and semantic consistency than on parameter count alone. A full description of this ablation study, along with detailed analysis and examples, is provided in Appendix C3.

---

### Note · Authors · 2025-12-30

I have read and agree with the venue's withdrawal policy on behalf of myself and my co-authors.